# Learning action-oriented models through active inference

**Alexander Tschantz**[1,2]*, **Anil K. Seth**[1,2,3], **Christopher L. Buckley**[2,4]*

**1** Sackler Centre for Consciousness Science, University of Sussex, Falmer, Brighton, United Kingdom,
**2** Department of Informatics, University of Sussex, Brighton, United Kingdom, **3** Canadian Institute for
Advanced Research, Azrieli Programme on Brain, Mind, and Consciousness, Toronto, Ontario, Canada,
**4** Evolutionary and Adaptive Systems Research Group, University of Sussex, Falmer, United Kingdom

* tschantz.alec@gmail.com (AT); c.l.buckley@sussex.ac.uk (CLB)

Learning action-oriented models through active
inference. PLoS Comput Biol 16(4): e1007805.

Irvine, UNITED STATES

**Data Availability Statement:** The Python code
used to simulate the models and the simulation
data are available at https://github.com/alec-
tschantz/action-oriented.

**Funding:** AT is funded by a PhD studentship from
the Sackler Foundation and the School of

## Abstract

Converging theories suggest that organisms learn and exploit probabilistic models of their
environment. However, it remains unclear how such models can be learned in practice.
The open-ended complexity of natural environments means that it is generally infeasible
for organisms to model their environment comprehensively. Alternatively, *action-oriented*
models attempt to encode a parsimonious representation of adaptive agent-environment
interactions. One approach to learning action-oriented models is to learn online in the
presence of goal-directed behaviours. This constrains an agent to behaviourally relevant
trajectories, reducing the diversity of the data a model need account for. Unfortunately,
this approach can cause models to prematurely converge to sub-optimal solutions,
through a process we refer to as a *bad-bootstrap*. Here, we exploit the normative frame-
work of *active inference* to show that efficient action-oriented models can be learned by
balancing goal-oriented and epistemic (*information-seeking*) behaviours in a principled
manner. We illustrate our approach using a simple agent-based model of bacterial chemo-
taxis. We first demonstrate that learning via goal-directed behaviour indeed constrains
models to behaviorally relevant aspects of the environment, but that this approach is
prone to sub-optimal convergence. We then demonstrate that epistemic behaviours facili-
tate the construction of accurate and comprehensive models, but that these models are
not tailored to any specific behavioural niche and are therefore less efficient in their use of
data. Finally, we show that active inference agents learn models that are parsimonious,
tailored to action, and which avoid bad bootstraps and sub-optimal convergence. Criti-
cally, our results indicate that models learned through active inference can support
adaptive behaviour in spite of, and indeed *because of*, their departure from veridical repre-
sentations of the environment. Our approach provides a principled method for learning
adaptive models from limited interactions with an environment, highlighting a route to
sample efficient learning algorithms.

Engineering and Informatics at the University of Sussex. CLB is supported by BBRSC grant number BB/P022197/1. We are grateful to the Dr. Mortimer and Theresa Sackler Foundation, which supports the Sackler Centre for Consciousness Science. AKS is additionally grateful to the Canadian Institute for Advanced Research (Azrieli Programme on Brain, Mind, and Consciousness). The funders had no role in study design, data collection and analysis, decision to publish, or preparation of the manuscript.

**Competing interests:** The authors have declared that no competing interests exist.

## Author summary

Within the popular framework of 'active inference', organisms learn internal models of their environments and use the models to guide goal-directed behaviour. A challenge for this framework is to explain how such models can be learned in practice, given (i) the rich complexity of natural environments, and (ii) the circular dependence of model learning and sensory sampling, which may lead to behaviourally suboptimal models being learned. Here, we develop an approach in which organisms selectively model those aspects of the environment that are relevant for acting in a goal-directed manner. Learning such 'action-oriented' models requires that agents balance information-seeking and goal-directed actions in a principled manner, such that both learning and information seeking are contextualised by goals. Using a combination of theory and simulation modelling, we show that this approach allows simple but effective models to be learned from relatively few interactions with the environment. Crucially, our results suggest that action-oriented models can support adaptive behaviour in spite of, and indeed because of, their departure from accurate representations of the environment.

## Introduction

In order to survive, biological organisms must be able to efficiently adapt to and navigate in their environment. Converging research in neuroscience, biology, and machine learning suggests that organisms achieve this feat by exploiting probabilistic models of their world [1–8]. These models encode statistical representations of the states and contingencies in an environment and agent-environment interactions. Such models plausibly endow organisms with several advantages. For instance, probabilistic models can be used to perform perceptual inference, implement anticipatory control, overcome sensory noise and delays, and generalize existing knowledge to new tasks and environments. While encoding a probabilistic model can be advantageous in these and other ways, natural environments are extremely complex and it is infeasible to model them in their entirety. Thus it is unclear how organisms with limited resources could exploit probabilistic models in rich and complex environments.

One approach to this problem is for organisms to selectively model their world in a way that supports action [9–14]. We refer to such models as *action-oriented*, as their functional purpose is to enable adaptive behaviour, rather than to represent the world in a complete or accurate manner. An action-oriented representation of the world can depart from a veridical representation in a number of ways. First, because only a subset of the states and contingencies in an environment will be relevant for behaviour, action-oriented models need not exhaustively model their environment [11]. Moreover, specific *misrepresentations* may prove to be useful for action [15–18], indicating that action-oriented models need not be accurate. By reducing the need for models to be isomorphic with their environment, an action-oriented approach can increase the tractability of the model learning process [19–24], especially for organisms with limited resources.

Within an action-oriented approach, an open question is how action-oriented models can be learned from experience. The environment, in and of itself, provides no distinction between states and contingencies that are relevant for behaviour and those which are not. However, organisms do not receive information passively. Rather, organisms *actively* sample information from their environment, a process which plays an important role in both perception and learning [23, 25–27]. One way that active sampling can facilitate the learning of efficient action-oriented models is to learn online in the presence of *goal-directed* actions. Performing

**A  Goal-directed cycle of learning & control**

1. Update model based on **goal-directed** observation

**Model**

4. Generate **goal-directed** observation

2. Use model to determine **goal-directed** action

**Environment**

3. Transition in **goal-directed** manner

**B  Maladaptive cycle of learning & control**

1. Update model based on **sub-optimal** observation

**Model**

4. Generate **sub-optimal** observation

2. Use model to determine **sub-optimal** action

**Environment**

3. Transition in **sub-optimal** manner

**C**  Observations with random actions

Optimal portion of data space

**D**  Observations with (sub-optimal) goal-directed actions

**E**  Observations with (optimal) goal-directed actions

**Fig 1. The coupling of learning and control. (A) Goal-directed cycle of learning and control.** A schematic overview of the coupling between a model and its environment when learning takes place in the presence of goal-directed actions. Here, a model is learned based on sampled observations. This model is then used to determine goal-directed actions, causing goal-relevant transitions in the environment, which in turn generate goal-relevant observations. **(B) Maladaptive cycle of learning and control.** A schematic overview of the model-environment coupling when learning in the presence of goal-directed actions, but for the case where a maladaptive model has been initially learned. The feedback inherent in the online learning scheme means that the model samples sub-optimal observations, which are subsequently used to update the model, thus entrenching maladaptive cycles of learning and control (bad bootstraps). **(C) Observations sampled from random actions.** The spread of observations covers the space of possible observations uniformly, meaning that a model of these observations must account for a diverse and distributed set of data, increasing the model's complexity. The red circle in the upper right quadrant indicates the region of observation space associated with optimal behaviour, which is only sparsely sampled. Note these are taken from a fictive simulation and are purely illustrative. **(D) Observations sampled from sub-optimal goal-directed actions.** Only a small portion of observation space is sampled. A model of this data would, therefore, be more parsimonious in its representation of the environment. However, the model prescribes actions that cause the agent to selectively sample a sub-optimal region of observation space (i.e outside the red circle in the upper-right quadrant). As the agent only samples this portion of observation space, the model does not learn about more optimal behaviours. **(E) Observations sampled from optimal goal-directed actions.** Here, as in **D**, the goal-directed nature of action ensures that only a small portion of observation space is sampled. However, unlike **D**, this portion is associated with optimal behaviours.

goal-directed actions restricts an organism to behaviourally relevant trajectories through an environment. This, in turn, structures sensory data in a behaviorally relevant way, thereby reducing the diversity and dimensionality of the sampled data (see Fig 1). Therefore, this approach offers an effective mechanism for learning parsimonious models that are tailored to an organism's adaptive requirements [19, 20, 23, 24, 28, 29].

Learning probabilistic models to optimise behaviour has been extensively explored in the model-based reinforcement learning (RL) literature [8, 30–32]. A significant drawback to existing methods is that they tend to prematurely converge to sub-optimal solutions [33]. One reason this occurs is due to the inherent coupling between action-selection and model learning. At the onset of learning, agents must learn from limited data, and this can lead to models that initially overfit the environment and, as a consequence, make sub-optimal predictions about the consequences of action. Subsequently using these models to determine goal-oriented actions can result in biased and sub-optimal samples from the environment, further

compounding the model's inefficiencies, and ultimately entrenching maladaptive cycles of learning and control, a process we refer to as a "bad-bootstrap" (see Fig 1).

One obvious approach to resolving this problem is for an organism to perform some actions, during learning, that are not explicitly goal-oriented. For example, heuristic methods, such as $\varepsilon$-greedy [34], utilise noise to enable exploration at the start of learning. However, random exploration of this sort is likely to be inefficient in rich and complex environments. In such environments, a more powerful method is to utilize the uncertainty quantified by probabilistic models to determine *epistemic* (or *intrinsic, information-seeking, uncertainty reducing*) actions that attempt to minimize the model uncertainty in a directed manner [35–40]. While epistemic actions can help avoid bad-bootstraps and sub-optimal convergence, such actions necessarily increase the diversity and dimensionality of sampled data, thus sacrificing the benefits afforded by learning in the presence of goal-directed actions. Thus, a principled and pragmatic method is needed to learn action-oriented models in the presence of both goal-directed *and* epistemic actions.

In this paper, we develop an effective method for learning action-oriented models. This method balances goal-directed and epistemic actions in a principled manner, thereby ensuring that an agent's model is tailored to goal-relevant aspects of the environment, while also ensuring that epistemic actions are contextualized by and directed towards an agent's adaptive requirements. To achieve this, we exploit the theoretical framework of active inference, a normative theory of perception, learning and action [41–43]. Active inference proposes that organisms maintain and update a probabilistic model of their typical (habitable) environment and that the states of an organism change to maximize the evidence for this model. Crucially, both goal-oriented and epistemic actions are complementary components of a single imperative to maximize model evidence—and are therefore evaluated in a common (information-theoretic) currency [38, 40, 43].

We illustrate this approach with a simple agent-based model of bacterial chemotaxis. This model is not presented as a biologically-plausible account of chemotaxis, but instead, is used as a relatively simple behaviour to evaluate the hypothesis that adaptive action-oriented models can be learned via active inference. First, we confirm that learning in the presence of goal-directed actions leads to parsimonious models that are tailored to specific behavioural niches. Next, we demonstrate that learning in the presence of goal-directed actions *alone* can cause agents to engage in maladaptive cycles of learning and control—'bad bootstraps'—leading to premature convergence on sub-optimal solutions. We then show that learning in the presence of epistemic actions allows agents to learn accurate and exhaustive models of their environment, but that the learned models are not tailored to any behavioural niche, and are therefore inefficient and unlikely to scale to complex environments. Finally, we demonstrate that balancing goal-directed and epistemic actions through active inference provides an effective method for learning efficient action-oriented models that avoid maladaptive patterns of learning and control. 'Active inference' agents learn well-adapted models from a relatively limited number of agent-environment interactions and do so in a way that benefits from systematic representational inaccuracies. Our results indicate that probabilistic models can support adaptive behaviour in spite of, and moreover, *because of*, the fact they depart from veridical representations of the external environment.

The structure of the paper is as follows. In section two, we outline the active inference formalism, with a particular focus on how it prescribes both goal-directed and epistemic behaviour. In section three, we present the results of our agent-based simulations, and in section four, we discuss these results and outline some broader implications. In section five, we outline the methods used in our simulations, which are based on the Partially Observed Markov

Decision Process (POMDP) framework, a popular method for modelling choice behaviour under uncertainty.

## Results

### Formalism

Active inference is a normative theory that unifies perception, action and learning under a single imperative—the minimization of variational *free energy* [42, 43]. Free energy $\mathcal{F}(\phi, o)$ is defined as:

$$
\begin{aligned}
\mathcal{F}(\phi, o) \quad &= \mathbb{KL}[Q(x|\phi)||P(x, o)] \\
&= \mathbb{KL}[Q(x|\phi)||P(x|o)] - \ln P(o)
\end{aligned}
\tag{1}
$$

where $\mathbb{KL}$ is the Kullback-Libeler divergence (KL-divergence) between two probability distributions, both of which are parameterized by the internal states of an agent. The first is the approximate posterior distribution, $Q(x|\phi)$, often referred to as the *recognition* distribution, which is a distribution over unknown or latent variables $x$ with sufficient statistics $\phi$. This distribution encodes an agent's 'beliefs' about the unknown variables $x$. Here, the term 'belief' does not necessarily refer to beliefs in the cognitive sense but instead implies a probabilistic representation of unknown variables. The second distribution is the generative model, $P(x, o)$, which is the joint distribution over unknown variables $x$ and observations $o$. This distribution encodes an agent's probabilistic model of its (internal and external) environment. We provide two additional re-arrangements of Eq 1 in Appendix 1.

Minimizing free energy has two functional consequences. First, it minimizes the divergence between the approximate posterior distribution $Q(x|\phi)$ and the true posterior distribution $P(x|o)$, thereby implementing a tractable form of approximate Bayesian inference known as variational Bayes [44]. On this view, perception can be understood as the process of maintaining and updating beliefs about hidden state variables $s$, where $s \in \mathcal{S}$. The hidden state variables can either be a compressed representation of the potentially high-dimensional observations (i.e. representing an object), or they can represent quantities that are not directly observable (i.e. velocity). This casts perception as a process of approximate inference, connecting active inference to influential theories such as the Bayesian brain hypothesis [45, 46] and predictive coding [47]. Under active inference, *learning* can also be understood as a process of approximate inference [43]. This can be formalized by assuming that agents maintain and update beliefs over the parameters $\theta$ of their generative model, where $\theta \in \Theta$. Finally, action can be cast as a process of approximate inference by assuming that agents maintain and update beliefs over control states $u$, where $u \in \mathcal{U}$, which prescribe actions $a$, where $a \in \mathcal{A}$. The delineation of control states from actions helps highlight the fact that actions are something which occur 'in the world', whereas control states are unknown random variables that the agent must infer. Together, this implies that $x = (s, \theta, u)$. Approximate inference, encompassing perception, action, and learning, can then be achieved through the following scheme:

$$
\phi^* \quad = \arg \min_{\phi} \mathcal{F}(\phi, o)
\tag{2}
$$

In other words, as new observations are sampled, the sufficient statistics $\phi$ are updated in order to minimize free energy (see the Methods section for the implementation used in the current simulations, or [48] for an alternative implementation based on the Laplace approximation). Once the optimal sufficient statistics $\phi^*$ have been identified, the approximate posterior will become an approximation of the true posterior distribution $Q(x|\phi^*) \approx P(x|o)$, meaning

that agents will encode approximately optimal beliefs over hidden states $s$, model parameters $\theta$ and control states $u$.

The second consequence of minimizing free energy is that it maximizes the Bayesian *evidence* for an agents generative model, or equivalently, minimizes 'surprisal' $-\ln P(o)$, which is the information-theoretic *surprise* of sampled observations (see Appendix 1). Active inference proposes that an agent's goals, preferences and desires are encoded in the generative model as a prior preference for favourable observations (e.g. blood temperature at 37˚) [49]. In other words, it proposes that an agent's generative model is biased towards favourable states of affairs. These prior preferences could be learned from experience, or alternatively, acquired through processes operating on evolutionary timescales. The process of actively minimizing free energy will, therefore, ensure that these favourable (i.e. probable) observations are preferentially sampled [50]. However, model evidence cannot be directly maximized through the inference scheme described by Eq 2, as the marginal probability of observations $P(o)$ is independent of the sufficient statistics $\phi$. Therefore, to maximize model evidence, agents must *act* in order to change their observations. This process can be achieved in a principled manner by selecting actions in order to minimize *expected* free energy, which is the free energy that is expected to occur from executing some (sequence of) actions [38, 51].

**Expected free energy.**   To ensure that actions minimize (the path integral of) free energy, an agent's generative model should specify that control states are *a-priori* more likely if they are expected to minimize free energy in the future, thus ensuring that the process of approximate inference assigns a higher posterior probability to the control states that are expected to minimize free energy [52]. The expected free energy for a candidate control state $\mathbf{G}_\tau(\phi_\tau, u_t)$ quantifies the free energy expected at some future time $\tau$ given the execution of some control state $u_t$, where $t$ is the current time point and:

$$
\begin{aligned}
\mathbf{G}_\tau(\phi_\tau, u_t) \quad &= \mathbb{E}_{Q(o_\tau, x_\tau | u_t, \phi_\tau)}[\ln Q(x_\tau | u_\tau, \phi_\tau) - \ln P(o_\tau, x_\tau | u_t)] \\
&\approx \underbrace{\mathbb{E}_{Q(o_\tau, x_\tau | u_t, \phi_\tau)}[\ln Q(x_\tau | u_t, \phi_\tau) - \ln Q(x_\tau | o_\tau, u_t, \phi_\tau)]}_{\text{(Negative) epistemic value}} \\
&\quad \underbrace{- \mathbb{E}_{Q(o_\tau, x_\tau | u_t, \phi_\tau)}[\ln P(o_\tau)]}_{\text{(Negative) instrumental value}}
\end{aligned}
\tag{3}
$$

We describe the formal relationship between free energy and expected free energy in Appendix 2. In order to evaluate expected free energy, agents must first evaluate the expected consequences of control, or formally, evaluate the predictive approximate posterior $Q(o_\tau, x_\tau | u_t, \phi_\tau)$. We refer readers to the Methods section for a description of this process.

The second (approximate) equality of Eq 3 demonstrates that expected free energy is composed of an *instrumental* (or *extrinsic, pragmatic, goal-directed*) component and an *epistemic* (or *intrinsic, uncertainty-reducing, information-seeking*) component. Note that under active inference, agents are mandated to *minimize* expected free energy, and as both the instrumental and epistemic terms are in a negative form in Eq 3, expected free energy will be minimized when instrumental and epistemic value are maximized. We provide a full derivation of the second equality in Appendix 3, but note here that the decomposition of expected free energy into instrumental and epistemic value affords an intuitive explanation. Namely, as free energy quantifies the divergence between an agent's current beliefs and its model of the world, this divergence can be minimized via two methods: by changing beliefs such that they align with observations (associated with maximizing epistemic value), or by changing observations such that they align with beliefs (associated with maximizing instrumental value).

Formally, instrumental value quantifies the degree to which the predicted observations $o_\tau$—given by the predictive approximate posterior $Q(o_\tau, x_\tau|u_t, \phi_\tau)$—are consistent with the agents prior beliefs $P(o_\tau)$. In other words, this term will be maximized when an agent expects to sample observations that are consistent with its prior beliefs. As an agent's generative model assigns a higher prior probability to favourable observations (i.e. goals and desires), maximizing instrumental value can be associated with promoting 'goal-directed' behaviours. This formalizes the notion that, under active inference, agents seek to maximize the evidence for their (biased) model of the world, rather than seeking to maximize reward as a separate construct (as in, e.g., reinforcement learning) [49].

Conversely, epistemic value quantifies the expected reduction in uncertainty in the beliefs over unknown variables $x$. Formally, it quantifies the expected information gain for the predictive approximate posterior $Q(x_\tau|u_t, \phi_\tau)$. By noting that that $x$ can be factorized into hidden states $s$ and model parameters $\theta$, we can rewrite *positive* epistemic value (i.e. the term to be maximized) as:

$$\underbrace{\mathbb{E}_{Q(o_\tau, s_\tau, \theta|u_t, \phi_\tau)}[\ln Q(s_\tau|o_\tau, u_t, \phi_\tau) - \ln Q(s_\tau|u_t, \phi_\tau)]}_{\text{State epistemic value}} +$$

$$\underbrace{\mathbb{E}_{Q(o_\tau, s_\tau, \theta|u_t, \phi_\tau)}[\ln Q(\theta|s_\tau, o_\tau, u_t, \phi_\tau) - \ln Q(\theta|\phi_\tau)]}_{\text{Parameter epistemic value}} \qquad (4)$$

We provide a full derivation of Eq 4 in Appendix 4 and discuss its relationship to several established formalisms. Here, we have decomposed epistemic value into *state* epistemic value, or *salience*, and *parameter* epistemic value, or *novelty*[53]. State epistemic value quantifies the degree to which the expected observations $o_\tau$ reduce the uncertainty in an agent's beliefs about the hidden states $s_\tau$. In contrast, parameter epistemic value quantifies the degree to which the expected observations $o_\tau$ and expected hidden states $s_\tau$ reduce the uncertainty in an agent's beliefs about model parameters $\theta$. Thus, by maintaining a distribution over model parameters, the uncertainty in an agent's generative model can be quantified, allowing for 'known unknowns' to be identified and subsequently acted upon [40]. Maximizing parameter epistemic value, therefore, causes agents to sample novel agent-environment interactions, promoting the exploration of the environment in a principled manner.

**Summary.** In summary, active inference proposes that agents learn and update a probabilistic model of their world, and act to maximize the evidence for this model. However, in contrast to previous 'perception-oriented' approaches to constructing probabilistic models [11], active inference requires an agent's model to be intrinsically biased towards certain (favourable) observations. Therefore, the goal is not necessarily to construct a model that accurately captures the true causal structure underlying observations, but is instead to learn a model that is tailored to a specific set of prior preferences, and thus tailored to a specific set of agent-environment interactions. Moreover, by ensuring that actions maximize evidence for a (biased) model of the world, active inference prescribes a trade-off between instrumental and epistemic actions. Crucially, the fact that actions are selected based on both instrumental *and* epistemic value means that epistemic foraging will be contextualized by an agent's prior preferences. Specifically, epistemic foraging will be biased towards parts of the environment that also provide instrumental value, as these parts will entail a lower expected free energy relative to those that provide no instrumental value. Moreover, the degree to which epistemic value determines the selection of actions will depend on instrumental value. Thus, when the instrumental value afforded by a set of actions is low, epistemic value will dominate action selection, whereas if actions afford a high degree of instrumental value, epistemic value will have less influence on the action selection. Finally, as agents maintain beliefs about (and thus quantify the uncertainty of) the hidden state

of the environment *and* the parameters of their generative model, epistemic value promotes agents to actively reduce the uncertainty in both of these beliefs.

## Simulation details

To test our hypothesis that acting to minimize expected free energy will lead to the learning of well-adapted action-oriented models, we empirically compare the types of model that are learned under four different action strategies. These are the (i) minimization of expected free energy, (ii) maximization of instrumental value, (iii) maximization of epistemic value, and (iv) random action selection, where the minimization of expected free energy (i) corresponds to a combination of the instrumental (ii) and epistemic (iii) strategies. For each strategy, we assess model performance after a range of model learning durations. We assess model performance across several criteria, including whether or not the models can prescribe well-adapted behaviour, the complexity and accuracy of the learned models, whether the models are tailored to a behavioural niche, and whether or not the models become entrenched in maladaptive cycles of learning and control ('bad-bootstraps').

We implement a simple agent-based model of bacterial chemotaxis that infers and learns based on the active inference scheme described above. Specifically, our model implements the 'adaptive gradient climbing' behaviour of *E. coli*. Note that we do not propose our model as a biologically realistic account of bacterial chemotaxis. Instead, we use chemotaxis as a relatively simple behaviour that permits a thorough analysis of the learned models. However, the active inference scheme described in this paper has a degree of biological plausibility [54], and there is some evidence to suggest that bacteria engage in model-based behaviours [55–58]. This behaviour depends on the chemical gradient at the bacteria's current orientation. In positive chemical gradients, bacteria 'run' forward in the direction of their current orientation. In negative chemical gradients, bacteria 'tumble', resulting in a new orientation being sampled. This behaviour, therefore, implements a rudimentary biased random-walk towards higher concentrations of chemicals. To simulate the adaptive gradient climbing behaviour of *E. coli*, we utilize the partially observed Markov Decision Process (POMDP) framework [59]. This framework implies that agents do not have direct access to the true state of the environment, that the state of the environment only depends on the previous state and the agent's previous action, and that all variables and time are discrete. Note that while agents operate on discrete representations of the environment, the true states of the environment (i.e the agent's position, the location of the chemical source, and the chemical concentrations) are continuous.

At each time step $t$, agents receive one of two observations, either a positive chemical gradient $o^{\text{pos}}$ or a negative chemical gradient $o^{\text{neg}}$. The chemical gradient is computed as a function of space (whether the agent is facing towards the chemical source) rather than time (whether the agent is moving towards the chemical source) [60], and thus only depends on the agent's current position and orientation, and the position of the chemical source. After receiving an observation, agents update their beliefs in order to minimize free energy. In the current simulations, agents maintain and update beliefs over three variables. The first is the hidden state variable $s$, which represents the agent's belief about the local chemical gradient, and which has a domain of $\{s^{\text{pos}}, s^{\text{neg}}\}$, representing positive and negative chemical gradients, respectively. The second belief is over the parameters $\theta$ of the agent's generative model, which describe the probability of transitions in the environment, given action. The final belief is over the control variable $u$, which has the domain of $\{u^{\text{run}}, u^{\text{tumble}}\}$, representing running and tumbling respectively. Agents are also endowed with the prior belief that observing positive chemical gradients $o^{\text{pos}}$ is *a-priori* more likely, such that the evidence for an agent's model is maximized (and free energy minimized) when sampling positive chemical gradients.

Once beliefs have been updated, agents execute one of two actions, either run $a^{\texttt{run}}$ or tumble $a^{\texttt{tumble}}$, depending on which of the corresponding control states was inferred to be more likely. Running causes the agent to move forward one unit in the direction of their current orientation, whereas tumbling causes the agent to sample a new orientation at random. The environment is then updated and a new time step begins. We refer the reader to the Methods section for a full description of the agents generative model, approximate posterior, and the corresponding update equations for inference, learning and action.

## Agents

All of the action strategies we compare infer posterior beliefs over hidden states, model parameters and control states via the minimization of free energy. However, they differ in how they assign prior (and thus posterior) probability to control states. The first strategy we consider is based on the minimization of *expected free energy*, which entails the following prior over control states:

$$
\begin{aligned}
P_{\texttt{EFE}}(u_t) \quad &= \sigma\big( \, \mathbb{E}_{Q(o_\tau, s_\tau, \theta | u_t, \phi_\tau)}[\ln Q(\theta | s_\tau, o_\tau, u_\tau, \phi_\tau) - \ln Q(\theta | \phi_\tau)] \\
&+ \mathbb{E}_{Q(o_\tau, s_\tau, \theta | u_t, \phi_\tau)}[\ln P(o_\tau)]\big)
\end{aligned}
\tag{5}
$$

where $\sigma(\cdot)$ is the softmax function, which ensures that $P_{\texttt{EFE}}(u_t)$ is a valid distribution. The first term corresponds to *parameter* epistemic value, or 'novelty', and quantifies the amount of information the agent expects to gain about their (beliefs about their) model parameters $\theta$. The second term corresponds to instrumental value and quantifies the degree to which the expected observations conform to prior beliefs. Therefore, the expected free energy agent selects actions that are expected to result in probable ('favourable') observations, and that are expected to disclose maximal information about the consequences of action. Note that in the following simulations, agents have no uncertainty in their likelihood distribution, which describes the relationship between the hidden state variables *s* and the observations *o* (see Methods). As such, the expected free energy agent does not assign probability to control states based on state epistemic value. Formally, when there is no uncertainty in the likelihood distribution, state epistemic value reduces to the entropy of the predictive approximate posterior over *s*, see [38]. For simplicity, we have omitted this term from the current simulations.

The second strategy is the *instrumental*, or 'goal-directed', strategy, which utilizes the following prior over control states:

$$
P_{\texttt{Instrumental}}(u_t) \quad = \sigma(\mathbb{E}_{Q(o_\tau, s_\tau, \theta | u_t, \phi_\tau)}[\ln P(o_\tau)])
\tag{6}
$$

The instrumental agent, therefore, selects actions that are expected to give rise to favourable observations. The third strategy is the *epistemic*, or 'information-seeking', strategy, which is governed by the following prior over control states:

$$
P_{\texttt{Epistemic}}(u_t) = \sigma(\mathbb{E}_{Q(o_\tau, s_\tau, \theta | u_t, \phi_\tau)}[\ln Q(\theta | s_\tau, o_\tau, u_\tau, \phi_\tau) - \ln Q(\theta | \phi_\tau)])
\tag{7}
$$

The epistemic agent selects actions that are expected to disclose maximal information about model parameters. The final strategy is the *random* strategy, which assigns prior probability to actions at random. These models were chosen to explore the relative contributions of instrumental and epistemic value to model learning, and crucially, to understand their combined influence. We predict that, when acting to minimize expected free energy, agent's will engage in a form of goal-directed exploration that is biased by their prior preferences, leading to adaptive action-oriented models. In contrast, we expect that (i) the instrumental agent will

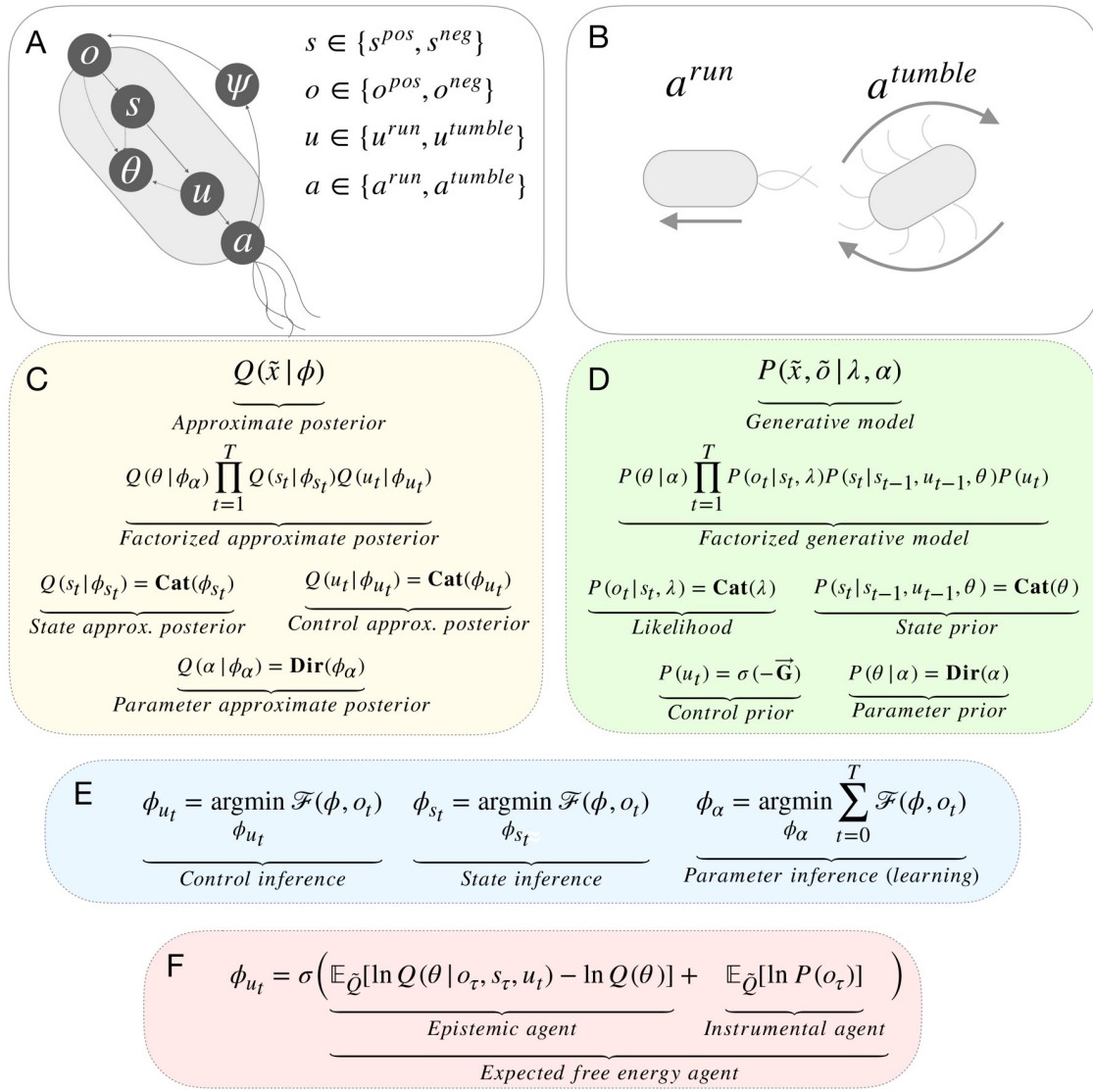

**Fig 2. Simulation & model details. (A) Agent overview.** Agents act in an environment which is described by states $\psi$, which are unknown to the agent but generate observations $o$. The agent maintains beliefs about the state of the environment $s$, however, $s$ and $\psi$ need not be homologous. Agents also maintain beliefs about control states $u$, which in turn prescribe actions $a$. Finally, the agent maintains beliefs over model parameters $\theta$, which describe the probability of transitions in $s$ under different control states $u$. **(B) Actions.** At each time step, agents can either *run*, which moves them forward one unit in the direction of their current orientation, or *tumble*, which causes a new orientation to be sampled at random. **(C) Approximate posterior.** The factorization of the approximate posterior, and the definition of each factor. In this figure, $x$ denotes the variables that an agent infers and $\phi$ denotes the parameters of the approximate posterior. We refer readers to Methods section for a full description of these distributions. **(D) Generative model.** The factorization of the generative model and the definition of each factor. Here, $\lambda$ denotes the parameters of likelihood distribution and $\alpha$ denotes the parameters of the prior distribution over parameters. We again refer readers to the methods section for full descriptions of these distributions. **(E) Free energy minimization.** The general scheme for free energy minimization under the mean-field assumption. We refer readers to the Methods section for further details. **(F) Control state inference.** The update equation for control state inference, where $\tilde{Q} = Q(o_\tau, s_\tau, \theta | u_t)$. This equation highlights the difference between the three action-strategies considered in the following simulations.

occasionally become entrenched in bad-bootstraps, due to the lack of exploration, and (ii) the epistemic agent will explore portions of state space irrelevant to behaviour, leading to slower learning. An overview of the model can be found in Fig 2 and implementation details for all four strategies are provided in the Methods section.

## Model performance

We first assess whether the learned models can successfully generate chemotactic behaviour. We quantify this by measuring an agent's distance from the source after an additional (i.e., post-learning) testing phase. Each testing phase begins by placing an agent at a random location and orientation 400 units from the chemical source. The agent is then left to act in the environment for 1000 time steps, utilizing the model that was learned during the preceding learning phase. No additional learning takes place during the testing phase. As the epistemic and random action strategies do not assign any instrumental (goal-oriented) value to actions, there is no tendency for them to navigate towards the chemical source. Therefore, to ensure a fair comparison between action strategies, all agents select actions based on the minimization of expected free energy during the testing phase. This allows us to assess whether the epistemic and random strategies can learn models that can support chemotactic behaviour, and ensures that any observed differences are determined solely by attributes of the learned models.

Fig 3a shows the final distance from the source at the end of the testing phase, plotted against the duration of the preceding learning phase, and averaged over 300 learned models for each action strategy and learning duration. The final distance of the expected free energy, epistemic and random strategies decreases with the amount of time spent learning, meaning that these action strategies were able to learn models which support chemotactic behaviour. However, the instrumental strategy shows little improvement over baseline performance, irrespective of the amount of time spent learning. Note that the first learning period consists of zero learning steps, meaning that the corresponding distance gives the (averaged) baseline performance for a randomly initialized model. This is less than the initial distance (400 units) as some of the randomly initialized models can support chemotaxis without any learning. The final distance from the source for the expected free energy, epistemic and random agents is not zero due to the nature of the adaptive-hill climbing chemotaxis strategy, which causes agents to not to settle directly on the source, but instead navigate around its local vicinity. Models

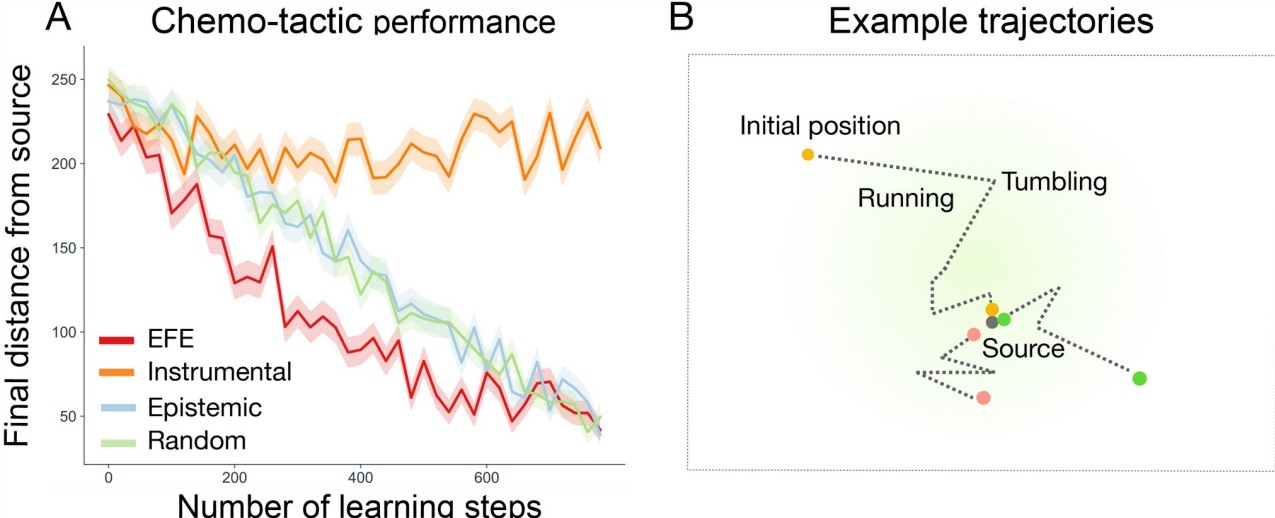

**Fig 3. (A) Chemotactic performance.** The average final distance from the chemical source after an additional testing phase, in which agents utilized the models learned in the corresponding learning phase. The average distance is plotted against the number of steps in the corresponding learning phase and is averaged over 300 models for each strategy and learning duration. Note that the *x*-axis denotes the number of time steps in the learning phase, rather than the number of time steps in the subsequent testing phase. Filled regions show +-SEM. **(B) Examples trajectories.** The spatial trajectories of agents who successfully navigated up the chemical gradient towards the chemical source.

learned by the expected free energy strategy consistently finish close to the chemical source, and learn chemotactic behaviour after fewer learning steps relative to the other strategies.

## Model accuracy

We now move on to consider whether learning in the presence of goal-oriented behaviour leads to models that are tailored to a behavioural niche. First, we assess how each action strategy affects the overall *accuracy* of the learned models. To test this, we measure the KL-divergence between the learned models and a 'true' model of agent-environment dynamics. Here, a 'true' model describes a model that has the same variables, structure and fixed parameters, but which has had infinite training data over all possible action-state contingencies. Due to the fact that the true generative process does not admit the notion of a prior, we measure the accuracy of the expectation of the approximate posterior distribution over parameters $\theta$, i.e. $\mathbb{E}[Q(\theta|\phi_\alpha)]$. Fig 4a shows the average accuracy of the learned models for each action strategy, plotted against the amount of time spent learning. These results demonstrate that the epistemic and random strategies consistently learn the most accurate models while the instrumental strategy consistently learns the least accurate models. However, the expected free energy strategy learns a model that is significantly less accurate than both the epistemic and random strategies, indicating that the most well-adapted models are not necessarily the most accurate.

Fig 4a additionally suggests that the epistemic and random strategies learn equally accurate models. This result may appear surprising, as the epistemic strategy actively seeks out transitions that are expected to improve model accuracy. However, given the limited number of possible state transitions in the current simulation, it is plausible that a random strategy offers a near-optimal solution to exploration. To confirm this, we evaluated the accuracy of models learned by the epistemic and random strategies in high-dimensional state space. The results of this experiment are given in Appendix 6, where it can be seen that the epistemic strategy does indeed learn models that are considerably more accurate than the random strategy.

We hypothesized that the expected free energy and instrumental strategies learned less accurate models because they were acting in a goal-oriented manner while learning. This, in turn, may have caused these strategies to selectively sample particular (behaviourally-relevant) transitions, at the cost of sampling other (behaviourally-irrelevant) transitions less frequently. To confirm this, we measured the distribution of state transitions sampled by each of the strategies after 1000 time steps learning, averaged over 300 agents. Because agents learn an *action-conditioned representation* of state transitions, i.e. $P(s_t|s_{t-1}, u_{t-1}, \theta)$, we separate state transitions that follow agents running from those that follow agents tumbling. Here, the notion of a state transition refers to a change in the state of the environment as a function of time, i.e. a positive to negative state transition implies that the agent was in a positive chemical gradient at time $t$ and a negative chemical gradient at $t + 1$. These results are shown in Fig 4b. For the epistemic and random strategies, the distribution is uniformly spread over (realizable) state transitions (running-induced transitions from positive to negative and negative to positive gradients are rare for all strategies, as such transitions can only occur in small portions of the environment). In contrast, the distributions sampled by the expected free energy and instrumental strategies are heavily biased towards a running-induced transitions from positive gradients to again a positive gradient. This is the transition that occurs when an agent is 'running up the chemical gradient', i.e., performing chemotaxis. The bias means that the remaining transitions between states are sampled less, relative to the epistemic and random strategies.

How do the learned models differ, among the four action strategies? To address this question, we measured the post-learning change in different distributions of the full model. This change reflects a measure of 'how much' an agent has learned about that particular

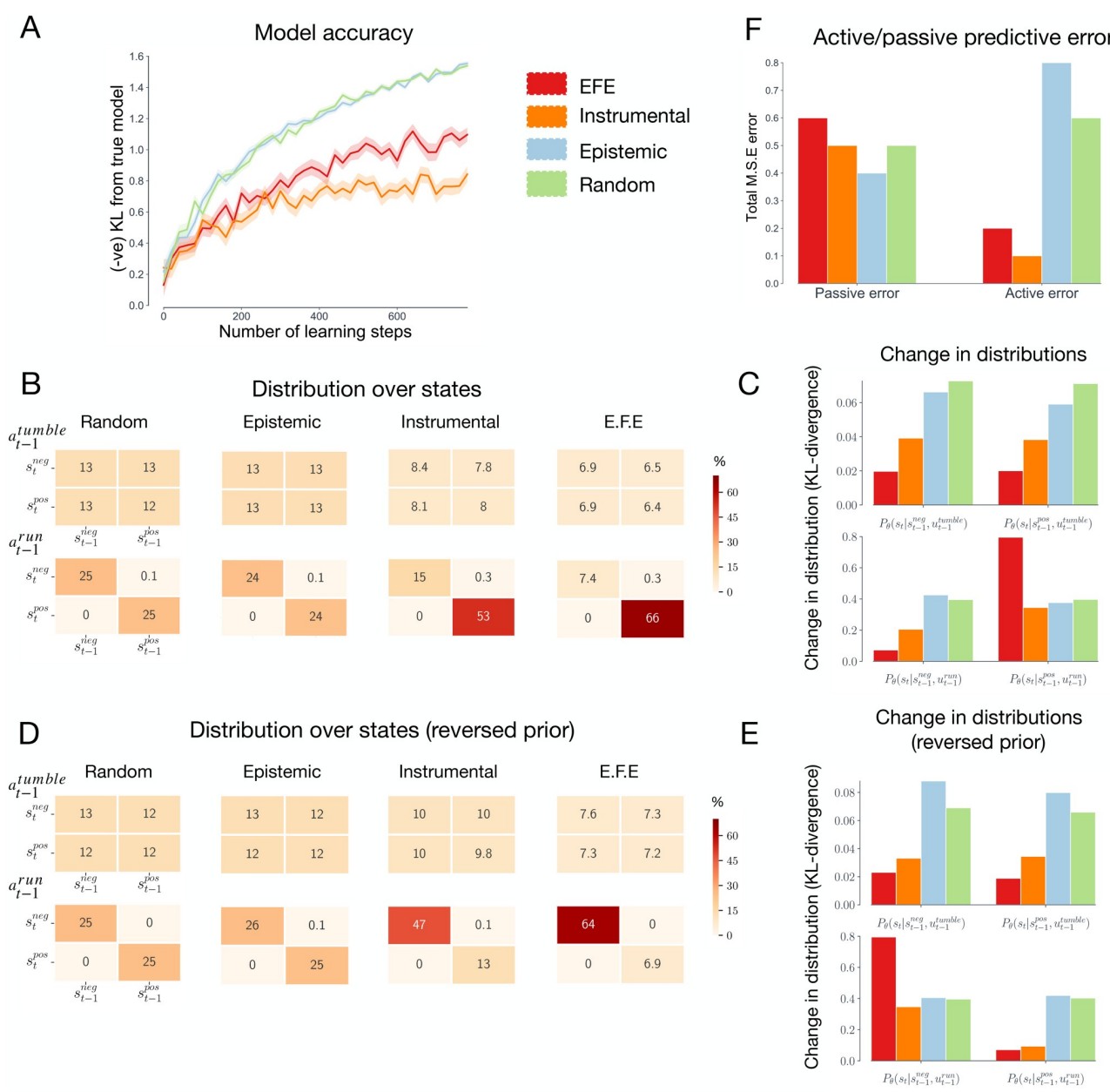

**Fig 4. Model accuracy. (A) Model accuracy.** The average *negative* model accuracy, measured as the KL-divergence from a 'true' model of agent-environment dynamics. The accuracy is plotted against the number of steps in the corresponding learning phase and is averaged over 300 models for each strategy. Filled regions show +-SEM. **(B) Distributions of state transitions.** The distribution of action-dependent state transitions for each strategy over 1000 learning steps, averaged over 300 models for each strategy. Here, columns indicate the state at the previous time step, whereas rows indicate the state following the transition. The top matrices display transitions that follow from tumbling, whereas the bottom matrices display transitions that follow from running. The numbers indicate the percentage of time that the corresponding state transition was encountered. For instance, the top left box denotes the percentage of time the agent experienced negative to negative state transitions following a tumbling action. Note that the distribution of transitions encountered by the epistemic and random strategies corresponds, within a small margin of error, to the distribution of transitions encountered by a 'true' model, i.e. a model that has been learned from infinite transitions with no behavioural biases. **(C) Change in distributions.** The average change in each of the distributions of the full learned model, measured as the KL-divergence between the original (randomly-initialized) distributions and the final (post-learning) distribution. Refer to Methods section for a description of these distributions. **(D & E) Reversed preferences.** These results are the same as for panels B & C, but for the case where agents have reversed preferences (i.e. priors). Here, agents believe running *down* chemical gradients to be more likely. The results demonstrate that the models of expected free energy and instrumental agent are sensitive to prior preferences. **(F) Active/passive prediction error.** The cumulative mean squared error of counterfactual predictions about state transitions, over 1000 steps learning and averaged over 300 agents. The active condition describes predictions of state-transitions following self-determined actions, whereas the passive condition describes predictions following random actions.

distribution. As described in the Methods, the full transition model $P(s_t|s_{t-1}, u_{t-1}, \theta)$ is composed of four separate categorical distributions. The first describes the effects of tumbling in negative gradients, the second describes the effects of tumbling in positive gradients, the third describes the effects of running in negative gradients, and fourth describes the effects of running in positive gradients. Fig 4c plots the KL-divergence between each of the original (randomly-initialized) distributions and the subsequent (post-learning) distributions. These results show that the expected free energy and instrumental strategies learn substantially less about three of the distributions, compared to the epistemic and random agents, explaining the overall reduction of accuracy displayed in Fig 4a. However, for the distribution describing the effects of running in positive gradients, the instrumental strategy learns as much as the epistemic and random strategies, while the expected free energy strategy learns substantially more. These results, therefore, demonstrate that acting in a goal-oriented manner biases an agent to preferentially sample particular (goal-relevant) transitions in the environment and that this, in turn, causes agents to learn more about these (goal-relevant) transitions.

To further verify this result, we repeated the analysis described in Fig 4b and 4c, but for the case where agents learn in the presence of reversed prior preferences (i.e. the agents believe that observing *negative* chemical gradients is *a-priori* more likely, and thus preferable). The results for these simulations are shown in 4d and 4e, where it can be seen that the expected free energy and instrumental strategy now preferentially sample running-induced transitions from negative to negative gradients, and learn more about the distribution describing the effects of running in negative gradients. This is the distribution relevant to navigating *down* the chemical gradient, a result that is expected if the learned models are biased towards prior preferences. By contrast, the models learned by the epistemic and random agents are not dependent on their prior beliefs or preferences.

## Active and passive accuracy

The previous results suggest that learning in the presence of goal-directed behaviour leads to models that are biased towards certain patterns of agent-environment interaction. To further elucidate this point, we distinguish between *active accuracy* and *passive accuracy*. We define active accuracy as the accuracy of a model in the presence of the agents own self-determined actions (i.e. the actions chosen according to the agent's strategy), and passive accuracy as the accuracy of a model in the presence of random actions. We measured both the passive and active accuracy of the models learned under different action strategies following 300 time-steps of learning. To do this, we let agents act in their environment for an additional 1000 time steps according to their action strategy, and, at each time step, measured the accuracy of their counterfactual predictions about state transitions. In the active condition, agents predicted the consequence of a self-determined action, whereas, in the passive condition, agents predicted the consequence of a randomly selected action. We then measured the mean squared error between the agents' predictions and the 'true' predictions (i.e. the predictions given by the 'true' model, as described for Fig 4a). The accumulated prediction errors for the passive and active conditions are shown in Fig 4f, averaged over 300 learned models for each strategy. As expected, there is no difference between the passive and active condition for the random strategy, as this strategy selects actions at random. The epistemic strategy shows the highest active error, which is due to the fact that the epistemic strategy seeks out novel (and thus less predictable) transitions. The instrumental strategy has the lowest active prediction error, and therefore the highest active accuracy. This is consistent with the view that learning in the presence of goal-directed behaviour allows agents to learn models that are accurate in the presence of their self-determined behaviour. Finally, the expected free energy strategy has an active error

that is lower than the epistemic and random strategies, but higher than the instrumental strategy. This arises from the fact that the expected free energy strategy balances both goal-directed and epistemic actions. Note that, in the current context, active accuracy is improved at the cost of passive accuracy. While the instrumental strategy learns the least accurate model, it is the most accurate at predicting the consequences of its self-determined actions.

## Pruning parameters

We now consider whether learning in the presence of goal-directed behaviour leads to *simpler* models of agent-environment dynamics. A principled way to approach this question is to ask whether each of the model's parameters are increasing or decreasing the Bayesian *evidence* for the overall model, which provides a measure of both the *accuracy* and the *complexity* of a model. In brief, if a parameter decreases model evidence, then removing—or 'pruning'—that parameter results in a model with higher evidence. This procedure can, therefore, provide a measure of how many 'redundant' parameters a model has, which, in turn, provides a measure of the complexity of a model (assuming that redundant parameters can, and should, be removed). We utilise the method of *Bayesian model reduction* [61] to evaluate the evidence for models with removed parameters. This procedure allows us to evaluate the evidence for reduced models without having to refit the model's parameters.

We first let each of the strategies learn a model for 500 time-steps. The parameters optimized during this learning period are then treated as priors for an additional (i.e., post-learning) testing phase. During this testing phase, agents act according to their respective strategies for an additional 500 time-steps, resulting in posterior estimates of the parameters.

Given the prior parameters $\alpha$ and posterior parameters $\phi_\alpha$, we can evaluate an approximation for the change in model evidence under a reduced model through the equation:

$$\Delta\mathcal{F} = \ln\mathbf{B}(\phi_\alpha) + \ln\mathbf{B}(\alpha') - \ln\mathbf{B}(\alpha) - \ln\mathbf{B}(\phi_\alpha + \alpha' - \alpha) \tag{8}$$

where $\ln\mathbf{B}(\cdot)$ is the beta function, $\alpha'$ are the prior parameters of the reduced model, and $\mathcal{F}$ is the variational free energy, which provides a tractable approximation of the Bayesian model evidence. See [40] for a derivation of Eq 8. If $\Delta\mathcal{F}$ is positive, then the reduced model—described by the reduced priors $\alpha'$—has less evidence than the full model, and *vice versa*. We remove each of the prior parameters individually by setting their value to zero and evaluate Eq 8. Fig 5a shows the percentage of trials that each parameter was pruned for each of the action strategies, averaged over 300 trials for each strategy. For the instrumental and epistemic agents, the parameters describing the effects of running in negative gradients and tumbling in positive gradients are most often pruned, as these are the parameters that are irrelevant to chemotaxis (which involves running in positive chemical gradients and tumbling in negative chemical gradients). In Fig 5b we plot the total number of parameters pruned, averaged over 300 agents. These results demonstrate that the expected free energy strategy entails models that have the highest number of redundant parameters, followed by the instrumental strategy. Under the assumption that redundant parameters can, and should, be pruned, the expected free energy and instrumental strategies learn simpler models, compared to the epistemic and random strategies. These results additionally suggest that pruning parameters will prove to be more beneficial (in terms of model complexity) for action-oriented models.

## Bad bootstraps and sub-optimal convergence

In the Introduction, we hypothesized that 'bad-bootstraps' occur when agents (and their models) become stuck in maladaptive cycles of learning and control, resulting in an eventual failure to learn well-adapted models. To test for the presence of bad-bootstraps, we allowed agents to

 

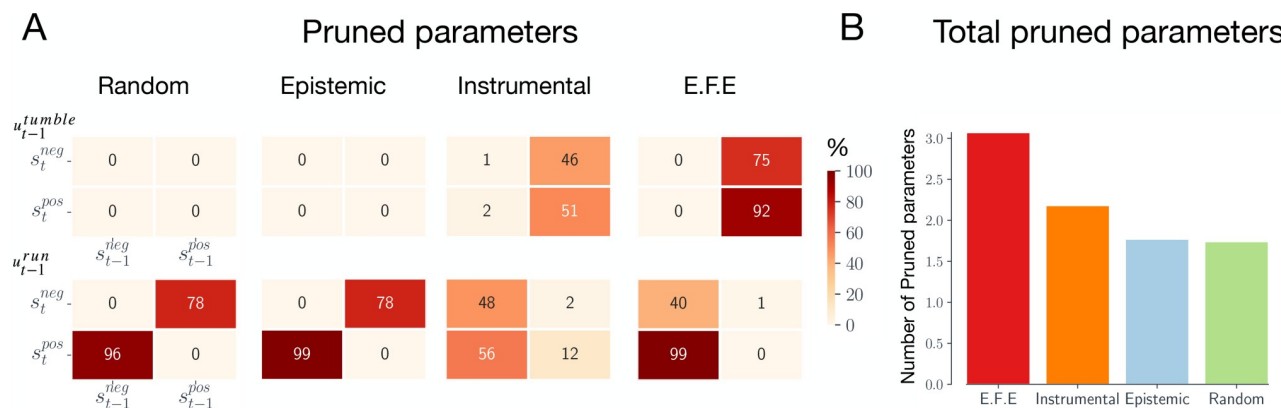

**Fig 5. Model complexity. (A) Number of pruned parameters.** Percentage of times each parameter was pruned, averaged over 300 agents. A parameter was pruned if it *decreased* the evidence for agents model. **(B) Total pruned parameters.** The average number of total number of pruned parameters, averaged over 300 agents.

learn models over an extended period of 4,000-time steps. We allowed this additional time to exclude the possibility that opportunities to learn had not been fully exploited by agents. (We additionally conducted the same experiment with 10,000-time steps; results were unchanged). We then tested the learned models on their ability to support chemotaxis, by allowing them to interact with their environment for an additional 1,000 time-steps using the expected free energy action strategy. To quantify whether the learned models were able to perform chemotaxis in any form, we measured whether the agent had moved more than 50 units towards the source by the end of the testing period.

After 4,000 learning steps, all the agents that had learned models using the expected free energy, epistemic or random strategies were able to perform at least some chemotaxis. In contrast 36% of the agents that had learned models under maximization of instrumental value did not engage in any chemotaxis at all. To better understand why instrumental agents frequently failed to learn well-adapted models, even after significant learning, we provide an analysis of a randomly selected failed model. This model prescribes a behavioural profile whereby agents continually tumble, even in positive chemical gradients. This arises from the belief that tumbling is more likely to give rise to positive gradients, even when the agent is in positive gradients. In other words, the model encodes the erroneous belief that, in positive gradients, running will be less likely to give rise to positive chemical gradients, relative to tumbling. Given this belief, the agent continually tumbles, and therefore never samples information that disconfirms this maladaptive belief. This exemplifies a 'bad bootstrap' arising from the goal-directed nature of the agent's action strategy.

Finally, we explore how assigning epistemic value to actions can help overcome bad bootstraps. We analyse an agent which acts to minimize expected free energy, quantifying the relative contributions of epistemic and instrumental value to running and tumbling. We initialize an agent with a randomly selected maladapted model and allow the agent to interact with (and learn from) the environment according to the expected free energy action strategy (i.e using the E.F.E agent). In Fig 6a, we plot the (negative) expected free energy of the running and tumbling control states over time, along with the relative contributions of instrumental and epistemic value. These results show that the (negative) expected free energy for the tumble control state is initially higher than that of the running control state because the agent believes there is less instrumental value in running. This causes the agent to tumble, which in turn causes the agent to gather information about the effects of tumbling. Consequently, the model becomes

 

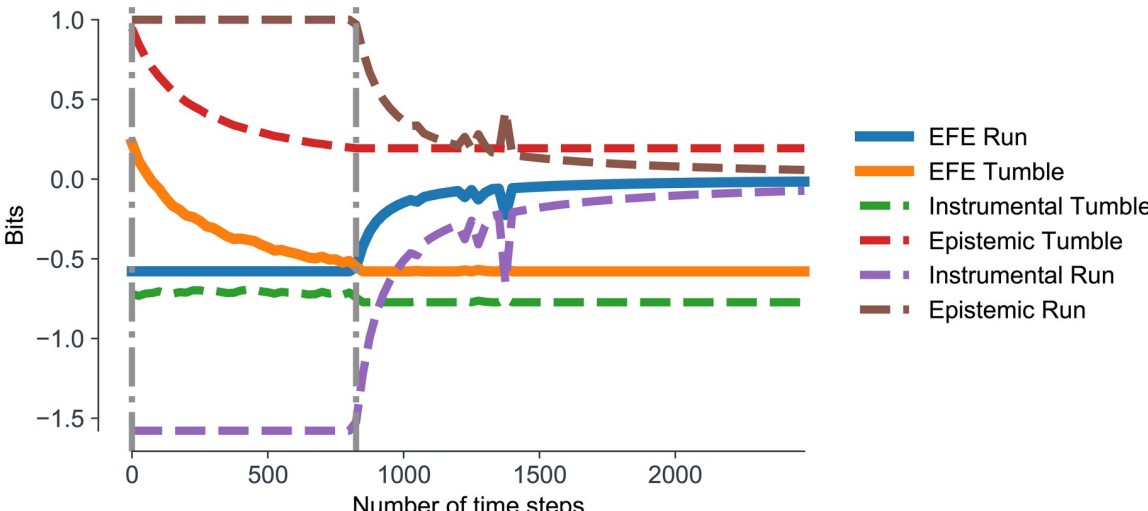

**Fig 6. Overcoming bad-bootstraps. (A) Expected free energy.** A plot of expected free energy for run and tumble control states overtime for an agent with an initially maladapted model. This model encodes the erroneous belief that running is less likely to give rise to positive chemical gradients, relative to tumbling. Therefore, at the start of the trial, the instrumental value of tumbling (green dotted line) is higher than the instrumental value of running (purple dotted line). The epistemic value of both running and tumbling (brown and red dotted lines, respectively) is initially the same. As the (negative) expected free energy for tumbling (orange line) is higher than the (negative) expected free energy for running (blue line), the agent tumbles for the first 900 time steps. During this time, agents gain information about the effects of tumbling, and the epistemic value of tumbling decreases, causing the negative expected free energy for tumbling to also decrease. This continues until the negative expected free energy is for tumbling is lower than the negative expected free energy for running, which has remained constant. Agents then run and gather information about the effects of running. This causes the epistemic value of running to decrease, but also causes the instrumental value of running to sharply increase, as the new information disconfirms their erroneous belief that running will not give rise to positive gradients.

less uncertain about the expected effects of tumbling, thereby decreasing the epistemic value of tumbling (and thus the (negative) expected free energy of tumbling). This continues until the negative expected free energy of tumbling becomes less than that of running, which has remained constant (since the agent has not yet gained any new information about running). At this point, the agent infers running to be the more likely action, which causes the agent to run. The epistemic value of running now starts to decrease, but as it does so the new sampled observations disclose information that running is very likely to cause transitions from positive to positive gradients (i.e., to maintain positive gradients). The instrumental value of running (and thus the negative expected free energy of running) therefore sharply increases in positive gradients, causing the agent to continue to run in positive gradients. Note that this agent did not fully resolve its uncertainty about tumbling. This highlights the fact that, under active inference, the epistemic value of an action is contextualized by current instrumental imperatives.

## Discussion

Equipping agents with generative models provides a powerful solution to prescribing well-adapted behaviour in structured environments. However, these models must, at least in part, be learned. For behaving agents—i.e., biological agents—the learning of generative models necessarily takes place in the presence of actions; i.e., in an 'online' fashion, during ongoing behaviour. Such models must also be geared towards prescribing actions that are useful for the agent. How to learn such 'action-oriented' models poses significant challenges for both computational biology and model-based reinforcement learning (RL).

In this paper, we have demonstrated that the active inference framework provides a principled and pragmatic approach to learning adaptive action-oriented models. Under this approach, the minimization of expected free energy prescribes an intrinsic and context-sensitive balance between goal-directed (instrumental) and information-seeking (epistemic) behaviours, thereby shaping the learning of the underlying generative models. After developing the formal framework, we illustrated its utility using a simple agent-based model of bacterial chemotaxis. We compared three situations. When agents learned solely in the presence of goal-directed actions, the learned models were specialized to the agent's behavioural niche but were prone to converging to sub-optimal solutions, due to the instantiation of 'bad-bootstraps'. Conversely, when agents learned solely in the presence of epistemic (information-seeking) actions, they learned accurate models which avoided sub-optimal convergence, but at the cost of reduced sample efficiency due to the lack of behavioural specialisation.

Finally, we showed that the minimisation of expected free-energy effectively-balanced goal-directed and information-seeking actions, and that the models learned in the presence of these actions were tailored to the agent's behaviours and goal, and were also robust to bad-bootstraps. Learning took place efficiently, requiring fewer interactions with the environment. The learned models were also less complex, relative to other strategies. Importantly, models learned via active inference departed in systematic ways from a veridical representation of the environment's true structure. For these agents, the learned models supported adaptive behaviour not only in spite of, but *because of*, their departure from veridicality.

## Learning action-oriented models: Good and bad bootstraps

When learning generative models online in the presence of actions, there is a circular dynamic in which learning is coupled to behaviour. The (partially) learned models are used to specify actions, and these actions provide new data which is then used to update the model. This circular dynamic (or 'information self-structuring' [20]) raises the potential for both 'good' and 'bad' bootstraps.

If actions are selected based purely on (expected) instrumental value, then the resulting learned models will be biased towards an agent's behavioural profile and goals (or prior preferences under the active inference framework—see Fig 4c & 4e), but will also be strongly constrained by the model's initial conditions. In our simulations, we showed that learning from instrumental actions was prone to the instantiation of 'bad-bootstraps'. Specifically, we demonstrated that these agents typically learned an initially maladapted model due to insufficient data or sub-optimal initialisation, and then subsequently used this model to determine goal-directed actions. This resulted in agents engaging with the environment in a sub-optimal and biased manner, thereby reintroducing sub-optimal data and causing models to become entrenched within local minima. Recent work in model-based RL has identified this coupling to be one of the major obstacles facing current model-based RL algorithms [62]. More generally, it is likely that bad-bootstraps are a prevalent phenomenon whenever parameters are used to determine the data from which the parameters are learned. Indeed, this problem played a significant role in motivating the (now common) use of 'experience replay' in model-free RL [63]. Experience replay describes the method of storing past experiences to be later sampled from for learning, thus breaking the tight coupling between learning and behaviour.

In the context of online learning, one way to avoid bad-bootstraps is to select actions based on (expected) epistemic value [37, 40, 53], where agents seek out novel interactions based on counterfactually informed beliefs about which actions will lead to informative transitions. By utilising the uncertainty encoded by (beliefs about) model parameters, this approach can proactively identify optimally informative transitions. In our simulations, we showed that agents

using this strategy learned models that asymptoted towards veridicality and, as such, were not tuned to any specific behavioural niche. This occurred because pure epistemic exploration treats all uncertainties as equally important, meaning that agents were driven to resolve uncertainty about all possible agent-environment contingencies. While models learned using this strategy were able to support chemotactic behaviour (Fig 3a), learning was highly sample-inefficient.

We have argued that a more suitable approach is to balance instrumental and epistemic actions in a principled way during learning. This is what is achieved by the active inference framework, via minimization of expected free energy. Minimizing expected free energy means that the model uncertainties associated with an agent's goals and desires are prioritised over those which are not. Furthermore, it means that model uncertainties are only resolved until an agent (believes that it) is sufficiently able to achieve its goals, such that agents need not resolve all of their model uncertainty. In our simulations, we showed that active inference agents learned models in a sample-efficient way, avoided being caught up in bad bootstraps, and generated well-adapted behaviour in our chemotaxis setting. Our data, therefore, support the hypothesis that learning via active inference provides a principled and pragmatic approach to the learning of well-adapted action-oriented generative models.

## Exploration vs. exploitation

Balancing epistemic and instrumental actions recalls the well-known trade-off between exploration and exploitation in reinforcement learning. In this context, the simplest formulation of this trade-off can be construed as a model-free notion in which exploration involves random actions. One example of this simple formulation is the *ε-greedy* algorithm which utilises noises in the action selection process to overcome premature sub-optimal convergence [34]. While an *ε-greedy* strategy might help overcome 'bad-bootstraps' by occasionally promoting exploratory actions, the undirected nature of random exploration is unlikely to scale to complex environments.

The balance between epistemic and instrumental actions in our active inference agents is more closely connected to the exploration-exploitation trade-off in model-based RL. As in our agents, model-based RL agents often employ exploratory actions that are selected to resolve model uncertainty. As we have noted, such actions can help avoid sub-optimal convergence (bad bootstraps), especially at the early stages of learning where data is sparse. However, in model-based RL it is normally assumed that, in the limit, a maximally comprehensive and maximally accurate (i.e., veridical) model would be optimal. This is exemplified by approaches that conduct an initial 'exploration' phase—in which the task is to construct a veridical model from as few samples as possible—followed by a subsequent 'exploitation' phase. By contrast, our approach highlights the importance of 'goal-directed exploration', in which the aim is not to resolve all uncertainty to construct a maximally accurate representation of the environment, but is instead to selectively resolve uncertainty until adaptive behaviour is (predicted to be) possible. Moreover, we have demonstrated that goal-directed exploration allows exploration to be contextualised by an agent's goals. Specifically, we have shown that acting to simultaneously explore and exploit the environment causes exploration to be biased towards parts of state space that are relevant for goal-directed behaviour, thereby increasing the efficiency of exploration. Therefore, our work suggests that acting to minimise expected free energy can benefit learning by naturally affording an efficient form of goal-directed exploration.

This kind of goal-directed exploration highlights an alternative perspective on the exploration-exploitation trade-off. We demonstrated that "exploitation"—traditionally associated with exploiting the agent's current knowledge to accumulate reward—can also lead to a type of

constrained learning that leads to 'action-oriented' representations of the environment. In other words, our results suggest that, in the context of model-learning, the "explore-exploit" dilemma additionally entails an "explore-constrain" dilemma. This is granted a formal interpretation under the active inference framework—as instrumental actions are associated with soliciting observations that are consistent with the model's prior expectations. However, given the formal relationship between instrumental value in active inference and the Bellman equations [43], a similar trade-off can be expected to arise in any model-based RL paradigm.

## Model non-veridicality

In our simulations, models learned through active inference were able to support adaptive behaviour even when their *structure* and *variables* departed significantly from an accurate representation of the environment. By design, the models utilized a severely impoverished representation of the environment. An exhaustive representation would have required models to encode information about the agent's position, orientation, the position of the chemical source, as well as a spatial map of the chemical concentrations so that determining an adaptive action would require a complex transformation of these variables. In contrast, our model was able to support adaptive behaviour by simply encoding a representation of the instantaneous effects of action on the local chemical gradient. Therefore, rather than encoding a rich and exhaustive internal mirror of nature, the model encoded a parsimonious representation of sensorimotor couplings that were relevant for enabling action [64]. While this particular 'action-oriented' representation was built-in through the design of the generative model, it nonetheless underlines that models need not be homologous with their environment if they are to support adaptive behaviour.

By evaluating the number of 'redundant' model parameters (as evaluated through Bayesian model reduction), we further demonstrated that learning in the presence of goal-directed behaviour leads to models that were more parsimonious in their representation of the environment, relative to other strategies (Fig 5b). Moreover, we showed that this strategy leads to models that did not asymptote to veridicality, in terms of the accuracy of the model's parameters (Fig 4a). Interestingly, these agents nevertheless displayed high 'active accuracy' (i.e., the predictive accuracy in the presence of self-determined actions), highlighting the importance of contextualising model accuracy in terms of an agent's actions and goals.

While these results demonstrate that models can support adaptive behaviour in spite of their misrepresentation of the environment and that these misrepresentations afforded benefits in terms of sample efficiency and model complexity, the active inference framework additionally provides a mechanism whereby misrepresentation *enables* adaptive behaviour. Active inference necessarily requires an organism's model to include systematic misrepresentations of the environment, by virtue of the organism's existence. Specifically, an organism's generative model must encode a set of prior beliefs that distinguish it from its external environment. For instance, the chemotaxis agents in the current simulation encoded the belief that observing positive chemical gradients was *a-priori* more likely. From an objective and passive point of view, these prior beliefs are, by definition, false. However, these systematic misrepresentations can be realized through action, thereby giving rise to apparently purposeful and autopoietic behaviour. Thus, under active inference, adaptive behaviour is achieved *because of*, and not just in spite of, a models departure from veridicality [15].

Encoding frugal and parsimonious models plausibly afford organism's several evolutionary advantages. First, the number of model parameters will likely correlate with the metabolic cost of that model. Moreover, simpler models will be quicker to deploy in the service of action and perception and will be less likely to overfit the environment. This perspective, therefore,

suggests that the degree to which exhaustive and accurate models are constructed should be mandated by the degree to which they are necessary for on-going survival. If the mapping between the external environment and allostatic responses is complex and manifold, then faithfully modelling features of the environment may pay dividends. However, in the case that frugal approximations and rough heuristics can be employed in the service of adaptive behaviour, such faithful modelling should be avoided. We showed that such "action-oriented" models arise naturally under ecologically valid learning conditions, namely, learning online in the presence of goal-directed behaviour. However, action-oriented behaviour that was adapted to the agent's goals only arose under the minimisation of expected free energy.

It is natural to ask whether there are scenarios in which action-oriented models might impede effective learning and adaptation. One such candidate scenario is transfer learning [65], whereby existing knowledge is reapplied to novel tasks or environments. This form of learning is likely to be important in biology, as for many organisms preferences can change over time. If the novel task or environment requires a pattern of sensorimotor coordination that is distinct from learned patterns of sensorimotor coordination, then a more exhaustive model of the environment might indeed facilitate transfer learning. However, if adaptation in the novel task or environment can be achieved through a subset of existing patterns of sensorimotor coordination (i.e. in going from walking to running), then one might expect an action-oriented representation to facilitate transfer learning, in so far as such representations reduce the search space for learning the new behaviour. This type of transfer learning is closely related to curriculum learning [66], whereby complex behaviours are learned progressively by first learning a series of simpler behaviours. We leave it to future work to explore the scenarios in which action-oriented models enable efficient transfer and curriculum learning.

## Active inference

While any approach to balancing exploration and exploitation is amenable to the benefits described in the previous sections, we have focused on the normative principle of active inference. From a purely theoretical perspective, active inference re-frames the exploration-exploitation dilemma by suggesting that both exploration and exploitation are complementary perspectives on a single objective function—the minimization of expected free energy. However, an open question remains as to whether this approach provides a practical solution to balancing exploration and exploitation. On the one hand, it provides a practically useful recipe by casting both epistemic and instrumental value in the same (information-theoretic) currency. However, the balance will necessarily depend on the shape of the agent's beliefs about hidden states, beliefs about model parameters, and prior beliefs about preferable observations. In the current work, we introduced an artificial weighting term to keep the epistemic and instrumental value within the same range. The same effect could have been achieved by constructing the shape (i.e. variance) of the prior preferences $P(o)$.

Active inference also provides a suitable framework for investigating the emergence of action-oriented models. Previous work has highlighted the fact that active inference is consistent with, and necessarily prescribes, frugal and parsimonious generative models, thus providing a potential bridge between 'representation-hungry' approaches to cognition espoused by classical cognitivism and the 'representation-free' approaches advocated by embodied and enactive approaches [6, 12, 13, 64, 67–75].

This perspective has been motivated by at least three reasons. First, active inference is proposed as a description of self-organization in complex systems [6]. Deploying generative models and minimizing free energy are construed as emergent features of a more fundamental drive towards survival. On this account, the purpose of representation is not to construct a

rich internal world model, but instead to capture the environmental regularities that allow the organism to act adaptively.

The second reason is that minimizing free energy implicitly penalizes the complexity of the generative model (see Appendix 1). This implies that minimizing free energy will reduce the complexity (or parameters) required to go from prior beliefs to (approximately) posterior beliefs, i.e. in explaining some observations. This occurs under the constraint of accuracy, which makes sure that the inferred variables can sufficiently account for the observations. In other words, minimizing free energy ensures that organism's maximize the accuracy of their predictions while minimizing the complexity of the models that are used to generate those predictions.

As discussed in the previous section, active inference also *requires* agents to encode systematic misrepresentations of their environment. Our work has additionally introduced a fourth motivation for linking active inference to adaptive action-oriented models, namely, that the minimization of expected free energy induces a balance between self-sustaining (and thus constrained) patterns of agent-environment interaction and goal-directed exploration.

The arguments and simulations presented in this paper resonate with previous work which views an active inference agent as a 'crooked scientist' [76, 77]. Here, an agent is seen as a 'scientist' insofar as it seeks out information to enable more accurate predictions. However, this work additionally highlights the fact that agents are biased by their own non-negotiable prior beliefs and preferences, leading them to seek out evidence for these hypotheses. We have built upon this previous work by exploring the types of models that are learned when an agent acts as a 'crooked scientist'.

## Conclusion

In this paper, we have demonstrated that the minimization of expected free energy (through active inference) provides a principled and pragmatic solution to learning action-oriented probabilistic models. These models can make the process of learning models of natural environments tractable, and provide a potential bridge between 'representation-hungry' approaches to cognition and those espoused by enactive and embodied disciplines. Moreover, we showed how learning online in the presence of behaviour can give rise to 'bad-bootstraps'—a phenomenon that has the potential to be problematic whenever learning is coupled with behaviour. Epistemic or information-seeking actions provide a plausible mechanism for overcoming bad-bootstraps. However, to exploration to be efficient, the epistemic value of actions must be contextualized by agents goals and desires. The ability to learn adapted models that are tailored to action provides a potential route to tractable and sample efficient learning algorithms in a variety of contexts, including computational biology and model-based RL.

## Methods

### The generative model

The agent's generative model specifies the joint probability over observations $o$, hidden state variables $s$, control variables $u$ and parameter variables $\theta$. To account for temporal dependencies among variables, we consider a generative model that is over a sequence of variables through time, i.e. $\tilde{x} = \{x_1, ..., x_t\}$, where tilde notation indicates a sequence from time $t = 0$ to the current time $t$, and $x_t$ denotes the value of $x$ at time $t$. The generative model is given by the

joint probability distribution $P(\tilde{o}, \tilde{s}, \tilde{u}, \theta | \lambda, \alpha)$, where:

$$P(\tilde{o}, \tilde{s}, \tilde{u}, \theta | \lambda, \alpha) = P(\theta | \alpha) \prod_{t=1}^{T} P(o_t | s_t, \lambda) P(s_t | s_{t-1}, u_{t-1}, \theta) P(u_t)$$

$$P(o_t | s_t, \lambda) = \mathbf{Cat}(\lambda)$$

$$P(s_t | s_{t-1}, u_{t-1}, \theta) = \mathbf{Cat}(\theta)$$

$$P(\theta | \alpha) = \mathbf{Dir}(\alpha)$$

$$P(u_t) = \sigma(-\tilde{\mathbf{G}})$$

(9)

where $\sigma(\cdot)$ is the softmax function. For simplicity, we initialize $P(s_{t=0})$ as a uniform distribution, and therefore exclude it from Eq 9.

The likelihood distribution specifies the probability of observing some chemical gradient $o_t$ given a belief about the chemical gradient $s_t$. This distribution is described by a set of categorical distributions, denoted $\mathbf{Cat}(\cdot)$, where each categorical distribution is a distribution over $k$ discrete and exclusive possibilities. The parameters of a categorical distribution can be represented as a vector with each entry describing the probability of some event $p_i$, with $\sum_{i=1}^{k} p_i = 1$. As the likelihood distribution is a conditional distribution, a separate categorical distribution is maintained for each hidden state in $\mathcal{S}$, (i.e. $s^{\text{pos}}$ and $s^{\text{neg}}$), where each of these distributions specifies the conditional probability of observing some chemical gradient (either $o^{\text{pos}}$ and $o^{\text{neg}}$). The parameters of the likelihood distribution can therefore be represented as a 2 x 2 matrix where each column $j$ is a categorical distribution that describes $P(o_t | s_t = j, \lambda)$. For the current simulations, we provide agents with the parameters $\lambda$ and do not require these parameters to be learned. The provided parameters encode the belief that there is an unambiguous mapping between $s^{\text{pos}}$ and $o^{\text{pos}}$, and between $s^{\text{neg}}$ and $o^{\text{neg}}$, meaning that $\lambda$ can be encoded as an identity matrix.

The prior probability over hidden states $s_t$ is given by the transition distribution $P(s_t | s_{t-1}, u_{t-1}, \theta)$, which specifies the probability of the current hidden state, given beliefs about the previous hidden state and the previous control state. In other words, this distribution describes an agent's beliefs about how running and tumbling will cause changes in the chemical gradient. Following previous work [38], we assume that agents know which control state was executed at the previous time step. As with the likelihood distribution, the prior distribution is described by a set of categorical distributions. Each categorical distribution $j$ specifies the probability distribution $P(s_t | s_{t-1} = j, \theta)$, such that $P(s_t | s_{t-1}, \theta)$ can again be represented as a 2 x 2 matrix. However, the transition distribution is also conditioned on control states $u$, meaning a separate transition matrix is maintained for both $u^{\text{run}}$ and $u^{\text{tumble}}$, such that the transition distribution can be represented as a 2 x 2 x 2 tensor. Agents, therefore, maintain separate beliefs about how the environment is likely to change for each control state.

We require agents to learn the parameters $\theta$ of the transition distribution. At the start of each learning period, we randomly initialize $\theta$, such that agents start out with random beliefs about how actions cause transitions in the chemical gradient. To enable these parameters to be learned, the generative model encodes (time-invariant) prior beliefs over $\theta$ in the distribution $P(\theta | \alpha)$. This distribution is modelled as Dirichlet distribution, denoted $\mathbf{Dir}(\cdot)$, where $\alpha$ are the parameters of this distribution. A Dirichlet distribution represents a distribution *over* the parameters of a distribution. In other words, sampling from this distribution returns a vector

of parameters, rather than a scalar. By maintaining a distribution over $\theta$, the task of learning about the environment is transformed into a task of inferring unknown variables.

Finally, the prior probability of control states is proportional to a softmax transformation of $-\tilde{\mathbf{G}}$, which is a vector of (negative) expected free energies, with one entry for each control state. This formalizes the notion that control states are *a-priori* more likely if they are expected to minimize free energy. We provide a full specification of expected free energy in the following sections.

## The approximate posterior

The approximate posterior encodes an agent's current approximately posterior beliefs about the chemical gradient $s$, the control state $u$ and model parameters $\theta$. As with the generative model, the approximate posterior is over a sequence of variables $Q(\tilde{s}, \tilde{u}, \theta|\phi)$, where $\phi$ are the sufficient statistics of the distribution.

In order to make inference tractable, we utilize the mean-field approximation to factorize the approximate posterior. This approximation treats a potentially high-dimensional distribution as a product of a number of simpler marginal distributions. Heuristically, this treats certain variables as statistically independent. Practically, it allows us to infer individual variables while keeping the remaining variables fixed. This approximation makes inference tractable, at the (potential) price of making inference sub-optimal. For inference to be optimal, the factorization of the approximate posterior must match the factorization of the true posterior.

Here, we factorize over time, the beliefs about the chemical gradient, the beliefs about model parameters and the beliefs about control states:

$$
\begin{aligned}
Q(\tilde{s}, \tilde{u}, \theta|\phi) &= Q(\theta|\phi_\alpha)\prod_{t=0}^{T}Q(s_t|\phi_{s_t})Q(u_t|\phi_{u_t}) \\
\\
Q(\theta|\phi_\alpha) &= \mathbf{Dir}(\phi_\alpha) \\
Q(s_t|\phi_{s_t}) &= \mathbf{Cat}(\phi_{s_t}) \\
Q(u_t|\phi_{u_t}) &= \mathbf{Cat}(\phi_{u_t})
\end{aligned}
\tag{10}
$$

## Inference, learning and action

Having defined the generative model and the approximate posterior, we can now specify how free energy can be minimized. In brief, this involves updating the sufficient statistics of the approximate posterior $\phi$ as new observations are sampled. To minimize free energy, we identify the derivative of free energy with respect to the sufficient statistics $\frac{\partial \mathcal{F}(\phi,o)}{\partial \phi}$, solve for zero, i.e. $\frac{\partial \mathcal{F}(\phi,o)}{\partial \phi} = 0$, and rearrange to give the variational updates that minimize free energy. Given the mean-field assumption, we can perform this scheme separately for each of the partitions of $\phi$, i.e $\phi_{s_t}$, $\phi_{u_t}$ and $\phi_\alpha$.

For the current scheme, the update equations for the hidden state parameters $\phi_s$ are (see Appendix 5 for a full derivation):

$$
\phi_{s_t} = \sigma(\ln P(o_t|s_t, \lambda) + \ln P(s_t|s_{t-1}, u_{t-1}, \theta))
\tag{11}
$$

This equation corresponds to state estimation or 'perception' and can be construed as a Bayesian filter that combines the likelihood of the current observation with a prior belief that is based on the previous hidden state and the previous control state. To implement this update in practice, we rewrite Eq 11 in terms of the relevant parameters and sufficient statistics (see Appendix 5):

$$\phi_{s_t} = \sigma(\ln\lambda \cdot \vec{o}_t + \bar{\theta}^{u_{t-1}} \cdot \phi_{s_{t-1}})$$

$$\bar{\theta}^{u_{t-1}} = \mathbb{E}_{Q(\theta|\phi_\alpha)}[\ln\theta^{u_{t-1}}] \tag{12}$$

$$= \psi(\phi_{\alpha_{ij}}^{u_{t-1}}) - \psi(\sum_{i=1}^{n}\phi_{\alpha_j}^{u_{t-1}})$$

Here, $\vec{o}_t$ is a one-hot encoded vector specifying the current observation, $\theta^u$ specifies the transition distribution corresponding to control state $u$, and $\psi(\cdot)$ is the digamma function. Note that the parameters of the likelihood distribution $\lambda$ are point-estimates of a categorical distribution, meaning it is possible to straightforwardly take the logarithm of this distribution. However, the beliefs about $\theta$ are described by the Dirichlet distribution $Q(\theta|\alpha)$, meaning that the mean of the logarithm of this distribution (denoted $\bar{\theta}$) must be evaluated (leading to lines two and three of Eq 12).

Learning can be conducted in a similar manner by updating the parameters $\phi_\alpha$ (see Appendix 5 for a full derivation):

$$\phi_\alpha^u = \alpha^u + \sum_{t=1}^{T}[a_{t-1} = u_{t-1}] \cdot \xi\phi_{s_t}\phi_{s_{t-1}} \tag{13}$$

where $[\cdot]$ is an inversion bracket that returns one if the statement inside the bracket is true and zero otherwise, and $\xi$ is an artificial learning rate, set to 0.001 for all simulations. Note that we update the parameters $\phi_\alpha$ after each iteration, but use a small learning rate to simulate the difference in time scales implied by the factorization of the generative model and approximate posterior. This update bears a resemblance to Hebbian plasticity, in the sense that the probability of each parameter increases if the corresponding transition is observed (i.e. 'fire together wire together').

Finally, actions can be inferred by updating the parameters $\phi_{u_t}$, where the update is given by (see Appendix 5 for a full derivation):

$$\phi_{u_t} = \sigma(-\tilde{\mathbf{G}}) \tag{14}$$

This equation demonstrates that the (approximately) posterior beliefs over control states are proportional to the vector of negative expected free energies. In other words, the posterior and prior beliefs about control states are identical.

## Expected free energy

In this section, we describe how to evaluate the vector $-\tilde{\mathbf{G}}$. This is a vector of negative expected free energies, with one for each control state $u \in \mathcal{U}$. As specified in the formalism, the negative expected free energy for a single control state is defined as $-\mathbf{G}_\tau(u_t)$, where $\tau$ is some future

time point, and, for the current simulations:

$$-\mathbf{G}_\tau(u_t) \quad = \underbrace{\mathbb{E}_{Q(o_\tau, s_\tau, \theta | u_t, \phi_\tau)}[\ln Q(\theta | s_\tau, o_\tau, u_t, \phi_\tau) - \ln Q(\theta | \phi_\tau)]}_{\text{Parameter epistemic value}}$$

$$+\underbrace{\mathbb{E}_{Q(o_\tau, s_\tau, \theta | u_t, \phi_\tau)}[\ln P(o_\tau)]}_{\text{Instrumental value}} \tag{15}$$

As described in the results section, we ignore the epistemic value for hidden states, as there is no uncertainty in the likelihood distribution. Moreover, for all simulations, $\tau = t + 1$, such that we only consider the immediate the immediate effects of action. This scheme is, however, entirely consistent with a sequence of actions, i.e. a policy.

In order to evaluate expected free energy, we rewrite Eq 15 in terms of parameters. By noting that $\mathbb{E}_{Q(o_\tau, s_\tau, \theta | u_t, \phi_\tau)}[\ln P(o_\tau)] = \mathbb{E}_{Q(o_\tau | u_t, \phi_\tau)}[\ln P(o_\tau)]$, we can write instrumental value as:

$$\mathbb{E}_{Q(o_\tau | u_t, \phi_\tau)}[\ln P(o_\tau)] = \phi_{o_\tau} \cdot \rho \tag{16}$$

where $\phi_{o_\tau}$ are the sufficient statistics of $Q(o_\tau | u_t, \phi_\tau)$, and $\rho$ are the parameters of $P(o_\tau)$, which is a categorical distribution, such that $\rho$ is a vector with one entry for each $o \in \mathcal{O}$. In order to evaluate parameter epistemic value, we utilise the following approximation:

$$\mathbb{E}_{Q(o_\tau, s_\tau, \theta | u_t, \phi_\tau)}[\ln Q(\theta | s_\tau, o_\tau, u_t, \phi_\tau) - \ln Q(\theta | \phi_\tau)] \quad \approx \phi_{s_\tau} \cdot \mathbf{W}^{u_t} \cdot \phi_{s_t}$$

$$\mathbf{W}^{u_t} \quad = \sum_{i=1}^{n} \phi_{\alpha_j}^{-1} - \phi_\alpha^{-1} \tag{17}$$

For details of this approximation, we refer the reader to [40]. For a given control state $u_t$, negative expected free energy can, therefore, be calculated as:

$$-\mathbf{G}_\tau(u_t) = \phi_{s_\tau} \cdot \mathbf{W}^{u_t} \cdot \phi_{s_t} + \delta(\phi_{o_\tau} \cdot \rho) \tag{18}$$

where $\phi_{s_\tau}$ are the sufficient statistics of $Q(s_\tau | u_t, \phi_\tau)$ and $\delta$ is an optional weighting term. For all simulations, this is set to 1/10. To calculate Eq 18, it is first necessary to evaluate the expected beliefs $Q(s_\tau | u_t, \phi_\tau)$ and $Q(o_\tau | u_t, \phi_\tau)$. The expected distribution over hidden states $Q(s_\tau | u_t, \phi_\tau)$ is given by $\mathbb{E}_{Q(s_t | u_t, \phi_\tau)}[P(s_\tau | s_t, u_t, \theta)]$. Given these beliefs over future hidden states, we can evaluate $Q(o_\tau | u_t, \phi_\tau)$ as $\mathbb{E}_{Q(s_t | u_t, \phi_\tau)}[P(o_\tau | s_\tau, \lambda)]$.

The full update scheme for the agents is provided in algorithm 1:

**Algorithm 1** Active inference MDP algorithm

```
Require: parameters of likelihood distribution λ, parameters of prior
  distribution over transition distribution parameters α, prior proba-
  bility of observations ρ
1: for t in T do
2:    o_t ← env.observe()              ◁ Sample observation from environment
3:    φ_{s_t} = σ(ln λ · o⃗_t + θ̄^{u_{t-1}} · φ_{s_{t-1}})        ◁ Hidden state
  inference
4:    φ_{u_t} = σ(−G⃗)                              ◁ Control state inference
5:    where − G_τ(u_t) = φ_{s_τ} · W^{u_t} · φ_{s_t} + φ_{o_τ} · ρ
                         └─ Epistemic agent ─┘   └ Instrumental agent ┘
                         └──── Expected free energy agent ────┘
6:    φ_α^u = α^u + Σ_{t=1}^T [a_{t-1} = u_{t-1}] · ξφ_{s_t}φ_{s_{t-1}}   ◁ Learning inference
7:    a_t ∼ Q(u_t | φ_{u_t})                         ◁ Sample action
```

```
8:   env.update(a_t)                              ◁ Perform action
9: end for
```

## Supporting information

**S1 Appendix. Rearrangements of the free energy functional.** In this appendix, we provide derivations for three arrangements of the free energy functional.
(PDF)

**S2 Appendix. Derivation of expected free energy.** In this appendix, we formally describe the relationship between free energy and expected free energy.
(PDF)

**S3 Appendix. Deriving instrumental and epistemic value.** In this appendix, we decompose expected free energy into instrumental and epistemic value.
(PDF)

**S4 Appendix. Relationship of epistemic value to established formalisms.** In this appendix, we demonstrate that epistemic value—a component of expected free energy—is equivalent to a number of established formalisms.
(PDF)

**S5 Appendix. Deriving the variational update equations for inference, learning and action.** In this appendix, we derive the update equations for beliefs about hidden states, control states and model parameters.
(PDF)

**S6 Appendix. Learning in a high-dimensional state space.** In this appendix, we present results for an additional experiment where we compare learning under epistemic and random action strategies in a high-dimensional state space.
(PDF)

## Author Contributions

**Conceptualization:** Alexander Tschantz, Anil K. Seth, Christopher L. Buckley.

**Data curation:** Alexander Tschantz.

**Formal analysis:** Alexander Tschantz.

**Investigation:** Alexander Tschantz, Christopher L. Buckley.

**Methodology:** Christopher L. Buckley.

**Supervision:** Anil K. Seth, Christopher L. Buckley.

**Writing – original draft:** Alexander Tschantz, Christopher L. Buckley.

**Writing – review & editing:** Alexander Tschantz, Anil K. Seth, Christopher L. Buckley.

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
