## [Decision Letter · Decision Letter 0]

11 Feb 2020

Dear Mr Tschantz,

Thank you very much for submitting your manuscript "Learning action-oriented models through active inference" for consideration at PLOS Computational Biology. As with all papers reviewed by the journal, your manuscript was reviewed by members of the editorial board and by several independent reviewers. The reviewers appreciated the attention to an important topic. Based on the reviews, we are likely to accept this manuscript for publication, providing that you modify the manuscript according to the review recommendations.

Sincerely,

Natalia L. Komarova

Deputy Editor

PLOS Computational Biology

Natalia Komarova

Deputy Editor

PLOS Computational Biology

[LINK]

Reviewer's Responses to Questions

**Comments to the Authors:**

Reviewer #1: This is a very interesting and well written article. It provides compelling theoretical arguments and computer simulations supporting the view that learning should balance instrumental and epistemic imperatives (as exemplified by active inference); and in favor of action-oriented models.

I have some comments on the theoretical part and the simulations, which I hope will help improving the manuscript.

Major comments

- The paper does not report how many control states are used for the simulation (nor the grid size, number of transition functions and parameters). By reading the paper I had the impression that only 2 control states were used. However, Appendix 6 mentions that the simulation is the same as the main text but with a bigger grid (15x15); and in this case 255 control states were used (which is somewhat odd, given that as at any moment the agent can select between run and tumble; is this correct?). Was the same method (one control state for each grid position) used also in the main simulation? And in that case, how did the agent know about its grid position? This point has also some theoretical implications as if so many control states are necessary, the idea that action-oriented models are more parsimonious than models encoding (for example) position is questionable.

- It would be interesting to discuss (or even better to simulate, but this is fully optional) how well learned models afford transfer learning. For example, how fast the different agents readapt to changes of prior preferences for gradients. This is important as for many biological organisms preferences can change over time.

- In the light of the author's discussion about exploration and exploitation, the most useful comparison would be with an "epsilon-greedy instrumental" agent. Is the epsilon-greedy mechanism (which is much simpler than epistemic exploration) sufficient to prevent bad bootstrap in instrumental agents? This could be discussed or (optionally) simulated.

Minor comments

- The authors cast the problem as POMDP. However, there is a one-to-one mapping between states and observations. Why not casting it as a MPD?

- The idea of representational inaccuracies is interesting. Is it fair to say that the inaccuracies that emerge in the transition functions are (just) ignorance, deriving from not having sampled some of the (infrequent or unselected) transitions?

- Page 7, around line 133. Please explain better the difference between control states and actions (and why the former are necessary in active inference but not in other frameworks), as these may sound confusing to most readers.

- Page 11, around line 117. The authors argue that "The goal is not, therefore, to construct a model that accurately captures the true causal structure underlying observations". This is correct, but it may be argued that "the goal is not necessarily" to do so. To the extent that knowing the true causal structure (or a close approximation) is useful, it could be learned by active inference agents.

- Page 11, around line 238. I think that for the sake of clarity the authors could reiterate that agent (i) is the "sum" of agents (ii) and (iii).

- Page 12, around line 268. Please explain "spatially rather than temporally"

- Figure 2 is interesting and useful. However, it (or its caption) should explicitly mention and explain all the parameters. There are some parameters like lambda and alpha that are not introduced in the caption.

- Figure 3 is well explained in the text. Despite so, it risks to be confusing (for distracted readers) as the text mentions that agents can learn for 1000 time steps and the x axis also reports an interval between 0 and 1000 (but it is about learning steps) - hence some readers may conflate the two and believe that the x axis reports time steps for action and not learning. Maybe you could simply change the interval in the x axis?

- The reported results seem to show that the overall performance of (all) agents is not excellent. Could you comment on that?

- Pag, 16, around line 354. The authors mention that they "measure the accuracy of the expectation of the approximate posterior distribution". Please clarify why you measure the expectation.

- The (interesting) results reported in figure 4 suggest that the EFE agent samples neg-neg transitions less often than the instrumental agent. Why this?

- In Figure 4B-D, it would be useful to show the distribution over states of the "true model" (the same used to measure model accuracy in Figure 4A. This would help making sense of some results; for example, the error of the instrumental agent is lower than the EFE agent; does this imply that the distributions it recovers (e.g., 53) are correct? Note also that these numbers are a bit difficult to interpret given that (as I mentioned above) things like the size of the grid etc. are not mentioned. Adding more details (and perhaps a figure) about the simulation scenario would make the results easier to interpret.

- Pag. 21, around line 485. The discussion of model complexity is interesting but could appear counterintuitive. After all, more parameters means more complexity. It may be useful to remark that pruning is necessary in in such kind of models?

- Pag 22, around line 519. It would be useful to mention that the agent used in the simulation is the EFE agent?

Reviewer #2: Review of: Learning action-oriented models through active inference

I have read this paper with great interest. It is one of the best computational papers on the free-energy principle I have seen. I would like to thank the authors for their clarity and thorough analysis of the system under scrutiny. I very much like the didactic nature of the paper. The argumentative steps, developed along with an explanation of the mathematical constructs, make for a very technical, yet highly readable paper. The chemotaxis model serves just as a toy model, exactly because of its simplicity it is able to make clear a number of potentially counterintuitive aspects of the behavior of expected-free-energy-minimizing agents.

I take the main result of the paper to be to show that EFE-agents learn qualitatively different and systematically biased models of their environment, which from a representational perspective are unexpected, but from an action-oriented and embodied perspective are less counterintuitive. Furthermore, learning is quite often unaddressed in the FEP literature, but plays a key role in this paper.

I have no major points of disagreement with the authors, see below a number of smaller points that I think could serve to improve the paper:

As an overall comment, some of the figures were very difficult to read (for example 4B). I could follow the narrative in the paper, but in these cases the figures were not very helpful.

p.8: The authors rightfully point out that active inference “proposes that an agent’s generative model is biased towards favourable states of affairs”. However, in the current simulation these “prior preferences” are taken as a given. Given that learning takes centre-stage in their paper, and given that the authors take all the other argumentative steps. Perhaps they can say something about how these preferences are acquired (learning seems not to be an option here, especially because of the bad-bootstrapping kind of scenarios the authors point out).

p.14: The authors introduce 4 learning strategies. At first I thought that the results of the paper were trivial, because, for example, an epistemic agent is explicitly not made to seek out the goal state. But since these strategies are only employed in the learning stage and not in the evaluation stage, this worry was unnecessary. Still, the authors might say something about their expectations in why specifically these models were chosen. And whether the results they obtained were anticipated.

p.17: “In contrast, the distributions sampled by the expected free energy and instrumental strategies are heavily biased towards a running-induced transition from positive to positive gradients. This is the transition that occurs when an agent is ‘running up the chemical gradient’, i.e., performing chemotaxis. The bias means that the remaining state transitions are sampled less, relative to the epistemic and random strategies.”

I found it hard to parse these few sentences. Not sure if the “positive” to “positive” is a mistake (or a different meaning of transition is implied here). It might be good to explain the idea of a state transition briefly in the paper (it bears the connotations of state transitions in physics, while what is implied here is a qualitative change in behaviour (i.e. tumbling or running).

p.26: I think the “goal-directed exploration” point, in contrast to more traditional solutions to the explore-exploit trade-off can be highlighted a bit more. Does EFE provide a more principle or integrated solution to this trade-off than more discrete ones (i.e. in which there is an irreversible switch from exploring to exploiting)?

I think the authors are very well on top of the literature, and I appreciate them engaging with the more theoretical/philosophical literature out there. I think the central idea of the paper can be very well be connected to what Bruineberg, Kiverstein and Rietveld (2016) call the “crooked scientist”. The EFE-agent as 1.) having non-negotiable preferences for the sensation it expects/wishes to encounter and 2) through this, exploring only that part of state-space that is relevant for its goal-directed activities. This second point really comes out nicely in the author’s paper and is a great continuation of the “crooked scientist” line of thinking. Although, the 2015 paper does not discuss expected free-energy, a follow up paper, Bruineberg et al., (2018) does discuss expected free-energy. I think the points made there (in the context of niche construction) are very much in line with and complementary to the current paper.

References:

Bruineberg, J., Kiverstein, J., & Rietveld, E. (2018). The anticipating brain is not a scientist: the free-energy principle from an ecological-enactive perspective. Synthese, 195(6), 2417-2444.

Bruineberg, J., Rietveld, E., Parr, T., van Maanen, L., & Friston, K. J. (2018). Free-energy minimization in joint agent-environment systems: A niche construction perspective. Journal of theoretical biology, 455, 161-178.

**Have all data underlying the figures and results presented in the manuscript been provided?**

Reviewer #1: Yes

Reviewer #2: Yes

PLOS authors have the option to publish the peer review history of their article (what does this mean?). If published, this will include your full peer review and any attached files.

Reviewer #1: No

Reviewer #2: No
---

## [Editor Report · Decision Letter 1]

19 Mar 2020

Dear Mr Tschantz,

We are pleased to inform you that your manuscript 'Learning action-oriented models through active inference' has been provisionally accepted for publication in PLOS Computational Biology.

Best regards,

Natalia L. Komarova

Deputy Editor

PLOS Computational Biology

Natalia Komarova

Deputy Editor

PLOS Computational Biology

---

## [Editor Report · Acceptance letter]

15 Apr 2020

PCOMPBIOL-D-19-01647R1 

Learning action-oriented models through active inference

Dear Dr Tschantz,

I am pleased to inform you that your manuscript has been formally accepted for publication in PLOS Computational Biology. Your manuscript is now with our production department and you will be notified of the publication date in due course.

With kind regards,

Matt Lyles
